# Knowledge, attitudes, and practices towards Human Papilloma Virus and uptake of HPV vaccine: A protocol for a systematic review

**Naharin Sultana Anni**[1]*, **Nadia Rehman**[2], **Agatha Nyambi**[2], **Anthony Musiwa**[2], **Tatyana Graham**[2], **Roseline Dzekem Dine**[2], **Maya Stevens-Uninsky**[1], **Elizabeth Alvarez**[2], **Zain Chagla**[3], **Laura Banfield**[4], **Lawrence Mbuagbaw**[2,5,6,7,8,9]

1 Department of Global Health, McMaster University, Hamilton, ON, Canada, 2 Department of Health Research Methods, Evidence and Impact, McMaster University, Hamilton, ON, Canada, 3 Department of Medicine, McMaster University, Hamilton, ON, Canada, 4 Health Sciences Library, McMaster University, Hamilton, ON, Canada, 5 Department of Anesthesia, McMaster University, Hamilton, ON, Canada, 6 Department of Pediatrics, McMaster University, Hamilton, ON, Canada, 7 Biostatistics Unit, Father Sean O'Sullivan Research Centre, St Joseph's Healthcare, Hamilton, ON, Canada, 8 Centre for Development of Best Practices in Health (CDBPH), Yaoundé Central Hospital, Yaoundé, Cameroon, 9 Division of Epidemiology and Biostatistics, Department of Global Health, Stellenbosch University, Cape Town, South Africa

* annin@mcmaster.ca

**Data Availability Statement:** All datasets supporting the conclusions of this paper are publicly accessible to readers. These datasets will

## Abstract

### Background

Despite a high burden of Human Papilloma Virus (HPV)-associated diseases, HPV vaccine uptake is disparate globally. The objective of this systematic review is to summarize the existing evidence on knowledge, attitudes, and practices (KAP) regarding HPV and the uptake of the HPV vaccine.

### Methods and analysis

We will conduct a systematic review of observational studies that report data on HPV KAP and vaccine uptake among people aged 16 and above. We will search MEDLINE, CINAHL, Embase, Emcare, Web of Science, Cochrane Library, Global Health, and PsycInfo. We will conduct screening, data extraction, and assessment of the methodological quality of the included studies in duplicate. A random-effects model will be used to pool data. Subgroup analysis will be done for age younger adults ($\leq$ 26 years old) and older adults (> 26 years old), sex (men and women), income level (as per World Bank), and WHO region. This systematic review will be reported according to the Preferred Reporting Items for Systematic Reviews and Meta-Analyses (PRISMA) guidelines. The PROSPERO registration number for the review is CRD42024532230.

### Ethics and dissemination

Ethical approval is not necessary as this study will review secondary published data. Our findings will be disseminated as part of a doctoral thesis and through peer-reviewed journal publications and conferences.

be provided either within the main manuscript of our systematic review or as additional supplementary information files whenever possible.

**Funding:** The author(s) received no specific funding for this work.

**Competing interests:** The authors have declared that no competing interests exist.

## Introduction

Human papillomavirus (HPV) is the most common sexually transmitted infection (STI) in the world, and it causes many cancers, including cervical, vulvar, anal, penile, and oropharyngeal cancer [1]. HPV types 16 and 18 are considered high-risk viruses, accounting for 70% of cervical cancer occurrences [2]. Cervical cancer is the second most prevalent cancer among women, with approximately 660,000 new cases and around 350,000 deaths occurring globally each year [3]. Men also experience rising HPV infection rates, similar to women. Nearly 1 in 3 men globally are infected with at least one genital HPV type, and around 1 in 5 men are infected with one or more high-risk (HR)-HPV types [4]. HPV-associated cancers are more prevalent in low- and middle income-countries (LMICs) compared to high-income countries (HICs), representing 6.7% of all cancers in LMICs and 2.8% in HICs [5, 6].

Despite the proven safety, efficacy, and cost-effectiveness of the HPV vaccine in men and women, only about 40% of LMICs implement vaccination, compared to over 80% of HICs [7]. HPV vaccines were not only introduced later in LMICs, but their rollout has also progressed at a slower pace due to challenges related to vaccine affordability, competing healthcare priorities, and resource constraints [8]. While international organizations like the World Health Organization (WHO), the Global Alliance for Vaccines and Immunization (GAVI) aid in HPV vaccine access, many upper and lower middle-income-countries don't meet eligibility criteria [9]. The WHO prioritizes HPV vaccination for girls aged 9 to 14, recognizing this age range as a critical window for vaccine effectiveness due to their lower likelihood of being sexually active [10]. Vaccination during this age range is equally important for boys, as it can prevent HPV transmission and related diseases, and helps in achieving herd immunity [2]. However, this policy leaves many girls, women, adolescent boys, and men outside this age range with limited or no access to the vaccine in many countries [10]. Conversely, individuals aged 16 and above, who are more likely to be sexually active, represent a demographic at higher risk for HPV infection [11]. In countries where males are included for HPV vaccination, adherence rates can be sub-optimal [12]. Some countries prioritize female HPV vaccination for herd immunity, neglecting at-risk males like men who have sex with men (MSM) and heterosexual men with unvaccinated partners [13]. Furthermore, the WHO recommends cervical cancer screening starting at age 30 for women, yet there are no equivalent screening protocols for men, which exacerbates the oversight of male populations who are also at risk for HPV-related diseases [2].

Lack of adequate knowledge about HPV and its vaccine is one of the most important barriers, both in HICs [14–16] and LMICs [7, 17]. Despite positive attitudes towards the HPV vaccine among various groups including adolescent girls and boys [18–21], parents [22, 23], college-aged men [13, 24] or women [25], men or women from racial/ethnic minorities [26, 27], MSM [28], and patients with immune-compromised conditions (e.g., AIDS, diabetes) [29], HPV vaccine uptake is low. Factors such as cost, incomplete insurance coverage and lack of health caregivers recommendations can contribute to this low vaccine uptake [30, 31]. Individual-level income may impact healthcare-seeking behavior and vaccine access, yet systemic barriers at the country level significantly shape HPV vaccine distribution and uptake [32]. In LMICs, specific challenges include the unavailability of the vaccine, lack of awareness, and exclusion of HPV vaccination from national immunization schedules [7, 9]. The availability of screening facilities, particularly in remote areas of LMICs, further limits efforts to prevent and manage HPV [9].

Geographical location plays a crucial role in knowledge, attitudes, and practices (KAP) towards HPV and HPV vaccine. In many African and Asian countries, cultural and religious beliefs, gender norms, and misconceptions about vaccine safety hinder HPV vaccination [33].

Open discussions about STIs are often lacking, affecting health education and HPV vaccine acceptance [9, 17, 33]. Despite high HPV-related disease prevalence, vaccine coverage in these regions falls below WHO targets because of these barriers [3]. In contrast, European and North American countries benefit from better healthcare infrastructure, higher awareness about HPV, and proactive health policies, resulting in higher HPV vaccine uptake [31, 34, 35]. However, misinformation and cultural stigma still pose significant barriers, with concerns about HPV vaccine safety, thereby causing progress to lag behind in fulfilling the WHO target [31, 32, 25].

To increase the uptake of HPV vaccines, it is important to conduct comprehensive systematic literature reviews that compare the KAP towards HPV and HPV vaccine among men and women, different age groups, different income level countries, and countries in different geographical location. To the best of our knowledge, there has been no systematic review or meta-analysis on this topic. Hence, the objective of this systematic review and meta-analysis is to summarise the evidence on KAP regarding HPV and the uptake of the HPV vaccine. Further, we will evaluate the association of country income, region, age, and gender with the differences and similarities of KAP towards HPV and its vaccine.

The research questions are-

1. What are the levels of KAP regarding HPV and its vaccine across different regions and income levels?

2. How do country income, region, age, and sex influence the uptake of the HPV vaccine and the variations in KAP towards HPV?

## Materials and methods

### Conceptual framework

This review is grounded in the Health Belief Model (HBM), which provides a conceptual framework for examining how individuals' beliefs about HPV susceptibility, severity, benefits of vaccination, and barriers to vaccination influence their knowledge, attitudes, and practices [36]. The HBM informs our approach to identifying key variables and interpreting the associations explored within this review.

This systematic review will be conducted and reported in accordance with the Preferred Reporting Items for Systematic Reviews and Meta-Analyses (PRISMA) [37] and (MOOSE) Meta-analysis and Systematic Reviews of Observational Studies guidelines [38]. The protocol for this meta-analysis has been registered with PROSPERO under the registration number CRD42024532230. We have attached a checklist as a S1 Table following PRISMA-P guidelines [39].

### Ethics and dissemination

Ethical approval is not necessary as this study will use secondary published data. Our findings will be disseminated as part of a doctoral thesis, peer-reviewed manuscripts, and at conferences.

### Inclusion and exclusion criteria

**Types of studies.** We will include observational studies (e.g. cohort studies, case-control studies, and cross-sectional studies) conducted between June 2006 and December 2024, as the HPV vaccine was licensed by the U.S. Food and Drug Administration (FDA) in June 2006 [1]. We will include studies for which the full text is available, regardless of the language of

publication. Reviews, editorials, and grey literature (dissertations, conference abstracts, and trial registries) will not be included.

**Types of participants.** We will include studies with data on individuals aged 16 and above, irrespective of sex or gender, who are presumed to be at risk for HPV and eligible for HPV vaccine. Studies with secondary sources of information (parents, caregivers, adolescents, healthcare providers, teachers, policymakers, programme administrators, managers, or other immunisation stakeholders) will not be included as our aim is to explore their KAP regarding HPV and its vaccine for themselves, independent of their roles as parents, caregivers, or healthcare providers.

**Types of outcomes.** The primary outcomes will be measured in this review include knowledge, attitudes, and practices.

1. **Knowledge:** Knowledge of HPV and the HPV vaccine refers to an individual's understanding and awareness of the virus, symptoms, mode of transmission, people at risk, treatments, and the available vaccine to prevent it [40, 41].

2. **Attitude:** Attitude toward HPV and the HPV vaccine refers to an individual's positive or negative evaluation of the virus and the vaccine, which can influence their decision to get vaccinated [40, 41].

3. **Practice:** Practice refers to behaviors aimed at preventing HPV, such as using condoms and engaging in safe sex, as well as actions related to receiving the recommended doses of the HPV vaccine [40, 41].

Secondary outcomes will assess how these KAP outcomes vary by demographic and contextual factors such as age, sex, income level, and geographic region. Specifically, we will explore if and how these variables are associated with KAP outcomes, examining whether and how such variations influence HPV vaccination practices and uptake.

## Information sources and search strategy

We will conduct a comprehensive literature search in MEDLINE, CINAHL, Embase, Web of Science Core Collection, Cochrane Library, Emcare, Global Health, and PsycInfo. We will use strategies customized to each database and their controlled vocabularies for medical subject headings (MeSH), keywords, and truncation search structures. The search will be based on these main concepts: *human papillomavirus*, *human papillomavirus vaccine*, *knowledge*, *attitude*, *and practice* under the guidance of a health sciences librarian at McMaster University. No language restrictions will be set.

Reference lists of review articles will be retrieved to identify additional sources of literature. A draft search strategy for MEDLINE has been attached in S2 Table.

## Data management

We will use EndNote X9, a bibliographic program [42], to store search results and delete duplicates. We will use DistillerSR for screening and data extraction. It is a web-based software management platform built for systematic reviews, to enhance data management and collaboration among team members [43]. Prior to the initial screening procedure, we will create and pre-test our title and abstract screening, full-text screening, and data extraction forms.

## Selection process

Reviewers (NSA, NR, AN, TG, AM, RDD, and MSU) will screen in pairs to independently the titles, abstracts, and full text of all retrieved articles to see if they meet the inclusion criteria.

Studies that are not eligible will be dropped. Each article will be reviewed by at least two reviewers working independently within their pairs, to ensure consistency. Any discrepancies or disagreements regarding inclusion will be resolved through discussion. If a consensus cannot be reached, a third-party adjudicator (e.g., LM) will make the final decision. We will create a PRISMA study flow diagram [39] that will display the number of articles identified, screened, included, and excluded, as well as the reasons for rejecting studies. Studies in languages other than English will be reviewed by colleagues at McMaster University with expertise in non-English languages through crowdsourcing within the global Cochrane community.

## Data extraction

A form designed specifically for this review will be used for data extraction. We will collect information on study characteristics, such as the first author's name, year of publication, contact information of the first author, country income level (according to the World Bank) [44], WHO geographical regions (African region, Region of the Americas, South-East Asia Region, European Region, Eastern Mediterranean Region and Western Pacific Region) [45], study design, sampling method, sample size, distribution of gender and age, study instrument (including questionnaires or scales to assess knowledge, attitude, or practice), methods of analysis, key findings (knowledge, attitude, and practice of HPV and uptake of HPV vaccine) and funding sources. Any factors associated with HPV KAP or HPV vaccine uptake will be collected. Data on HPV KAP and HPV vaccine uptake can be reported as counts (percent) or means (standard deviation), depending on the nature of the tool used. For binary data, we will collect the numerator and denominator. For continuous data, we will collect the mean, standard deviation (SD), and number of participants contributing to the estimate.

## Risk of bias in individual studies

We will use the Newcastle-Ottawa Scale (NOS) to assess the risk of bias in case-control and cohort studies [46]. The NOS has a total of 10 points across 3 domains: selection (maximum 5 points), comparability (maximum 3 points), and outcomes (2 points). For cross-sectional studies, we will use a modified version of the Newcastle-Ottawa Scale [47], which allocates 8 points across selection (maximum 4 points), comparability (maximum 2 points), and outcomes (2 points). To evaluate the risk of bias, articles will be categorized into three groups based on their scores: low risk (8–10 points for NOS and 6–8 points for the modified version), medium risk (5–7 points for NOS and 3–5 points for the modified version), and high risk (0–5 points for NOS and 0–2 points for the modified version). The quality of the studies will be assessed by two authors independently.

## Data synthesis

Data will first be categorized based on type (categorical vs. continuous) and then further organized by the type of questions asked. For example, proportions for the number of correct responses to similar questions will be pooled, as well as average scores on agreement statements when questions and statements are comparable [48–50].

To synthesize the primary outcomes, we will pool summary statistics for KAP, and HPV vaccine uptake separately, using forest plots to present pooled proportions with 95% confidence intervals (CI). Similarly, we will pool the means (SD) of KAP for HPV and the uptake of the HPV vaccine and present them in separate forest plots. This approach allows us to analyze each type of outcome appropriately and prevent the loss of valuable data.

We will conduct a meta-analysis using the RevMan 5 software (version 5.4.1, Cochrane Collaboration) with a random-effects model [51]. The reason for choosing this method, that it

demonstrates better properties in the presence of heterogeneity (if any), accounting for both within-study and between-study variances [52].

For categorical data reported as proportions, we will pool the estimates using random-effects meta-analysis of proportions. The variances will be stabilized using the Freeman-Tukey double arcsine transformation. Exact confidence intervals will be computed by inverting the equal-tailed test based on the binomial distribution. Pooled estimates will be computed using the Dersimonian and Laird method based on the transformed values and their variances.

For data reported as means and standard deviation, we will use the generic inverse variance approach to pool means. Given that they will likely be reported using different scales, we will standardize all the means, bringing them to the same scale before pooling. If data are reported as medians, they will be converted to medians using the approach proposed by Luo et al. [53]. The findings will be presented as forest plots.

## Investigation of statistical heterogeneity

We will assess statistical heterogeneity using the $\chi2$ test for homogeneity and the $I^2$ statistic to quantify inconsistency.

## Subgroup analysis

We will conduct subgroup analysis for age categorized as younger adults ($\leq$ 26 years old) and older adults (>26 years old) [54]; sex categorized as: men and women; between WHO regions and income levels (HICs and LMICs). Studies will be put in a specific age category if at least 80% of the total sample is within the age range of interest.

**Sensitivity analysis.** We will perform a sensitivity analysis to assess the robustness of the findings by examining the impact risk of bias and outliers.

## Assessment of reporting biases

If we have ten or more studies for our primary outcome, we will assess for asymmetry in the funnel plot to see if there is any publication bias [55]. We will also conduct Egger's test for small study effects [55].

## Certainty of evidence

We will assess the certainty of the evidence using Grading of Recommendations, Assessment, Development, and Evaluation (GRADE) [56]. The certainty of evidence for each outcome will be graded as high, moderate, low, or extremely low. We will look at the risk of bias, imprecision, inconsistency, indirectness, and publication bias [56].

## Discussion

This systematic review aims to comprehensively identify and synthesize the gaps in knowledge, attitudes, and practices related to HPV and the uptake of the HPV vaccine across the world. We will systematically highlight variations in KAP and vaccine uptake across different demographic and socio-economic variables that may be associated with them, including income levels and geographical regions within countries, as well as the age and gender of individuals. By elucidating these disparities, our findings will inform the development of targeted interventions designed to enhance public knowledge about HPV and its vaccine, improve attitudes toward vaccination, and subsequently influence positive vaccination practices.

To the best of our knowledge, this systematic review represents the first attempt to summarize the global evidence on KAP regarding HPV and the uptake of the HPV vaccine without

language restrictions. This study will be beneficial to new researchers in the field, provide an updated summary of the evidence and guide future studies where requires. Furthermore, the findings of the systematic review can potentially contribute to shaping public health policies and interventions aimed at improving the KAP of HPV, increasing the uptake of HPV vaccines, and reducing the burden of HPV-related diseases.

There will be some imitations as well. First, given the diverse nature of the studies to be included, substantial heterogeneity is anticipated, which may compromise our ability to pool data effectively. However, we will perform sensitivity analyses and subgroup analyses, to explore and account for potential sources of heterogeneity. In addition, most studies included in this review will likely be cross-sectional surveys and may be limited to low- and middle-income countries, which may potentially limit our ability to make robust comparisons. Hence, we will transparently discuss the implications of these limitations in our interpretation of the findings, thereby ensuring the validity and reliability of our conclusions.

## Supporting information

**S1 Table. PRISMA-P (Preferred Reporting Items for Systematic review and Meta-Analysis Protocols) 2015 checklist: Recommended items to address in a systematic review protocol.** (DOCX)

**S2 Table. Search strategy.** (DOCX)

## Acknowledgments

We acknowledge the support of Laura Banfield, information specialist from the McMaster University Health Sciences Library, for her expert assistance in search strategy development and search guidance.

## Author Contributions

**Conceptualization:** Naharin Sultana Anni, Lawrence Mbuagbaw.

**Data curation:** Lawrence Mbuagbaw.

**Investigation:** Naharin Sultana Anni, Lawrence Mbuagbaw.

**Methodology:** Naharin Sultana Anni, Lawrence Mbuagbaw.

**Project administration:** Naharin Sultana Anni, Lawrence Mbuagbaw.

**Resources:** Naharin Sultana Anni, Lawrence Mbuagbaw.

**Software:** Naharin Sultana Anni, Lawrence Mbuagbaw.

**Supervision:** Lawrence Mbuagbaw.

**Validation:** Lawrence Mbuagbaw.

**Visualization:** Naharin Sultana Anni, Lawrence Mbuagbaw.

**Writing – original draft:** Naharin Sultana Anni.

**Writing – review & editing:** Nadia Rehman, Agatha Nyambi, Anthony Musiwa, Tatyana Graham, Roseline Dzekem Dine, Maya Stevens-Uninsky, Elizabeth Alvarez, Zain Chagla, Laura Banfield, Lawrence Mbuagbaw.

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
