## [Decision Letter · Decision Letter 0]

15 Sep 2024

PONE-D-24-23976Knowledge, attitudes, and practices towards Human Papilloma Virus and uptake of HPV vaccine: a protocol for a systematic reviewPLOS ONE

Dear Dr. Anni,

Thank you for submitting your manuscript to PLOS ONE. After careful consideration, we feel that it has merit but does not fully meet PLOS ONE’s publication criteria as it currently stands. Therefore, we invite you to submit a revised version of the manuscript that addresses the points raised during the review process.

I appreciate the authors's effort in developing this systematic review protocol. The topic is interesting, and synthesizing the findings from these studies in a systematic review would be a valuable contribution to the literature. However, I am concerned about the potential challenges of pooling data from studies that use different questionnaires. The authors should establish a clear strategy for aggregating KAP data across the included studies. Simply combining KAP scores from different studies using varying questionnaires would not be appropriate. Additionally, pooling data from similar questions could be problematic; for instance, a seemingly straightforward knowledge question like "How many HPV vaccine doses are required?" might elicit different responses depending on how the question is framed in each study.

We look forward to receiving your revised manuscript.

Kind regards,

Hamidreza Karimi-Sari

Academic Editor

PLOS ONE

Journal Requirements:

“The authors have no competing interests to declare.”

Reviewers' comments:

Reviewer's Responses to Questions

**Comments to the Author**

1. Does the manuscript provide a valid rationale for the proposed study, with clearly identified and justified research questions?

Reviewer #1: Yes

Reviewer #2: Yes

2. Is the protocol technically sound and planned in a manner that will lead to a meaningful outcome and allow testing the stated hypotheses?

Reviewer #1: Partly

Reviewer #2: Yes

3. Is the methodology feasible and described in sufficient detail to allow the work to be replicable?

Reviewer #1: Yes

Reviewer #2: Yes

4. Have the authors described where all data underlying the findings will be made available when the study is complete?

Reviewer #1: No

Reviewer #2: Yes

5. Is the manuscript presented in an intelligible fashion and written in standard English?

Reviewer #1: Yes

Reviewer #2: Yes

6. Review Comments to the Author

You may also provide optional suggestions and comments to authors that they might find helpful in planning their study.

Reviewer #1: Introduction-

1. Please provide an updated prevalence & burden of HPV injection globally.

2. Mention the screening protocol as per the WHO.

3. Mentioned risk factors need to be modified ( eg- in LIC the risk factors may be lack of knowledge, unviability of the vaccine, cultural & religious beliefs, lack of screening facilities in the remote areas, not mentioning in Immunization schedule etc.

4. include the importance of vaccination for the male and age of vaccination.

Objectives

1. Mention the Key questions ( Objectives ) clearly

Methodology

1. Analytic frame work is not clear.

2. Exclusion criteria includes parents, teachers, health care professionals and inclusion criteria mentions as > 16yrs population, so it is confusing.

3. In "type of outcomes"- PRACTICE aspect needs to be modified.

Reviewer #2: This systematic review aims to synthesize evidence on KAP regarding HPV and HPV vaccine uptake. The authors plan to also identify factors that influence knowledge and awareness of HPV and its vaccine, as well as attitudes and practices regarding

HPV and HPV vaccination.

This is a well written protocol and I commend the authors. However, I have a few comments

1. Page 3, 2nd paragraph : "LMICs not only started being introduced later, but they progressed at a slower pace due to challenges in vaccine affordability, competing healthcare priorities, and resource constraints. " I am not sure of what was said here. I believe there was a typographical error somewhere.

2. Page 6 "Reviewers (NSA, NR, AN, TG, AM, RDD, and MSU) will screen the titles, abstracts, and full text of all retrieved articles independently and in duplicate to see if they meet the inclusion criteria. Studies that are not eligible will be dropped. The reviewers will address disagreements at any stage of selection by discussion or by adjudication by a third party". It is generally good practice to have (at least) 2 reviewers screen and perform full text review of each study then disagreements regarding each study are resolved by a third party. While this may be what the reviewers meant I don't think this was not clear in the sentence above. I think the authors should revise this to make their methodology for screening and full text review as clear as possible. The statement above may also be interpreted as each reviewer has their set of studies to review independently (ie all the studies will only be reviewed by one reviewer independently)

3. I am not sure how the authors plan to do a meta-analysis here. They rightfully pointed out that their outcomes are very broad concepts and different studies may have capture these differently. For instance, Study 1 reporting a composite knowledge score of 50% may be different from the composite score 50% that study 2 reports. This is true because what was used to measure knowledge of HPV or HPV vaccination may be entirely different for study 1 and 2. Do the authors plan to use the specific proportions reported eg XX% reported that they had heard of of HPV vaccine in study 1 and then get a pooled proportion if this was reported in study 2? If this is what they plan to do, I think it should be made clearer in their protocol.

7. PLOS authors have the option to publish the peer review history of their article (what does this mean?). If published, this will include your full peer review and any attached files.

Reviewer #1: **Yes: **Sophia Cyril Vincent

Reviewer #2: No

---

## [Author Response · Author response to Decision Letter 0]

31 Oct 2024

Comments from Editor and reviewers Responses from Authors 

Academic editor

I appreciate the authors’ effort in developing this systematic review protocol. The topic is interesting, and synthesizing the findings from these studies in a systematic review would be a valuable contribution to the literature. However, I am concerned about the potential challenges of pooling data from studies that use different questionnaires. The authors should establish a clear strategy for aggregating KAP data across the included studies. Simply combining KAP scores from different studies using varying questionnaires would not be appropriate. Additionally, pooling data from similar questions could be problematic; for instance, a seemingly straightforward knowledge question like "How many HPV vaccine doses are required?" might elicit different responses depending on how the question is framed in each study.

Thank you for your valuable feedback and for the opportunity to revise our manuscript. We appreciate your insights and have addressed the concerns raised to enhance the clarity and rigor of our approach to synthesizing Knowledge, Attitudes, and Practices (KAP) data across studies with different questionnaires.

We recognize the challenges inherent in synthesizing data from studies that may utilize diverse instruments and question structures.

We will only be pooling comparable questions across studies. Data will first be categorized based on the type of data: categorical vs continuous and then further organized by the type of questions asked. For example, proportions for the number of correct responses for similar questions can be pooled or the average score on agreement with a statement (provided the questions and statements are similar). This approach has been used for COVID KAP meta-analyses.1,2,3 While many studies use common validated items from the World Health Organization (WHO) or the CDC, they often must make adaptations to fit the cultural or demographic context of the target population.

We have revised the data synthesis part of our protocol. See Please see page 8, Data synthesis lines 15-23. 

Reviewer #1: 

 Thank you for your valuable comments. Your feedback helped us to improve the quality of our manuscript. We have addressed each of your comments as follows and provided more detail below to ensure ease of review. Based on your comment, we made changes in the introduction, objectives and methods sections.

Introduction-.

1. Please provide an updated prevalence & burden of HPV infection globally We have updated the manuscript based on the latest data published by the WHO and a recent systematic review (2023). Please see the introduction, page 3, lines 5-7 and 8-9.

“Cervical cancer is the second most prevalent cancer among women, with approximately 660,000 new cases and around 350,000 deaths occurring globally each year.”

“Nearly 1 in 3 men globally are infected with at least one genital HPV type, and around 1 in 5 men are infected with one or more high-risk (HR)-HPV types.”

Reference 3 and 4 have been added to support the new data. 

2. Mention the screening protocol as per the WHO. We have mentioned the WHO screening protocol. See the introduction page 3, lines 30-32.

“Furthermore, the WHO recommends cervical cancer screening starting at age 30 for women, yet there are no equivalent screening protocols for men, which exacerbates the oversight of male populations who are also at risk for HPV-related diseases.”

3. Mentioned risk factors need to be modified (eg- in LIC the risk factors may be lack of knowledge, unviability of the vaccine, cultural & religious beliefs, lack of screening facilities in the remote areas, not mentioning in Immunization schedule etc. In response to this comment, we have rearranged the whole paragraph and included this information.

See the introduction page 4, first two paragraphs and lines 10-13.

“In LMICs, specific challenges include the unavailability of the vaccine, lack of awareness, and exclusion of HPV vaccination from national immunization schedules. The availability of screening facilities, particularly in remote areas of LMICs, further limits efforts to prevent and manage HPV.”

4. include the importance of vaccination for the male and age of vaccination.

 We have now mentioned the importance of vaccination for men and age of vaccination.

See the introduction page 3, lines 12-13 and lines 21-23.

“Despite the proven safety, efficacy, and cost-effectiveness of the HPV vaccine in men and women, only about 40% of LMICs implement vaccination, compared to over 80% of HICs.”

“Vaccination during this age range is equally important for boys, as it can prevent HPV transmission and related diseases, and helps in achieving herd immunity.”

Objectives

1. Mention the Key questions (Objectives) clearly

 We updated the research questions. See the introduction page 5, lines 1- 5.

It now reads as: 

“1. What are the levels of KAP regarding HPV and its vaccine across different regions and income levels?

2. How do country income, region, age, and sex influence the uptake of the HPV vaccine and the variations in KAP towards HPV?”. 

Methodology

1. Analytic framework is not clear. We have clarified our analytical framework. 

Our review is grounded in Health Belief Model (HBM) which we have included in manuscript. See the conceptual framework page 5, lines 6-11.

“This review is grounded in the Health Belief Model (HBM), which provides a conceptual framework for examining how individuals' beliefs about HPV susceptibility, severity, benefits of vaccination, and barriers to vaccination influence their knowledge, attitudes, and practices. The HBM informs our approach to identifying key variables and interpreting the associations explored within this review.”

Reference 36 has been added to support the new data. 

Please see page 6, lines 7 and 18-21.

The primary outcomes measured in this review will include knowledge, attitudes, and practices related to HPV and HPV vaccination. Secondary outcomes will assess how these KAP outcomes vary by demographic and contextual factors such as age, sex, income level, and geographic region. Specifically, we will explore if and how these variables are associated with KAP outcomes, examining whether and how such variations influence HPV vaccination practices and uptake.

We also revised our data synthesis sections for clarifying the processes. Please see page 8, Data synthesis lines 15-23.

2. Exclusion criteria include parents, teachers, health care professionals and inclusion criteria mention as > 16yrs population, so it is confusing.

 We appreciate your feedback on the exclusion criteria. Our inclusion criteria focus on individuals aged 16 years and above who are presumed to be directly at risk of HPV and eligible for vaccination. As such, the exclusion criteria specifically exclude proxy sources of information, such as responses from parents, teachers, and healthcare professionals, who may not have accurate information about the individuals of interest. This approach helps ensure that the data reflects direct experiences and perspectives rather than second-hand knowledge or influence from roles as caregivers or educators. 

Please see materials and methods page 5, lines 28-29 and page 6, lines 1-5.

3. In "type of outcomes"- PRACTICE aspect needs to be modified. We refined the definition of practice. Please see page 6, lines 15-17.

“Practice refers to behaviors aimed at preventing HPV, such as using condoms and engaging in safe sex, as well as actions related to receiving the recommended doses of the HPV vaccine.”

Reviewer #2: 

This systematic review aims to synthesize evidence on KAP regarding HPV and HPV vaccine uptake. The authors plan to also identify factors that influence knowledge and awareness of HPV and its vaccine, as well as attitudes and practices regarding

HPV and HPV vaccination. This is a well written protocol, and I commend the authors. However, I have a few comments 

Thank you for your valuable feedback and for considering our manuscript for revision. We appreciate the opportunity to improve our work and carefully addressed the concerns raised.

1. Page 3, 2nd paragraph: "LMICs not only started being introduced later, but they progressed at a slower pace due to challenges in vaccine affordability, competing healthcare priorities, and resource constraints. " I am not sure of what was said here. I believe there was a typographical error somewhere. We rephrased the sentence to improve clarity. See the introduction page 3, lines 14-16.

“HPV vaccines were not only introduced later in LMICs, but their rollout has also progressed at a slower pace due to challenges related to vaccine affordability, competing healthcare priorities, and resource constraints.”

2. Page 6 "Reviewers (NSA, NR, AN, TG, AM, RDD, and MSU) will screen the titles, abstracts, and full text of all retrieved articles independently and in duplicate to see if they meet the inclusion criteria. Studies that are not eligible will be dropped. The reviewers will address disagreements at any stage of selection by discussion or by adjudication by a third party". It is generally good practice to have (at least) 2 reviewers screen and perform full text review of each study then disagreements regarding each study are resolved by a third party. While this may be what the reviewers meant I don't think this was not clear in the sentence above. I think the authors should revise this to make their methodology for screening and full text review as clear as possible. The statement above may also be interpreted as each reviewer has their set of studies to review independently (ie all the studies will only be reviewed by one reviewer independently) We have revised the methodology to specify that reviewers will work in pairs, with each study independently screened by two reviewers to enhance consistency and rigor in the selection process. Discrepancies within pairs will be resolved through discussion, and a third-party adjudicator will make the final decision if consensus is not reached. 

Please see the updated text in the “Selection Process” section on page 7, lines 9-14.

3. I am not sure how the authors plan to do a meta-analysis here. They rightfully pointed out that their outcomes are very broad concepts, and different studies may have capture these differently. For instance, Study 1 reporting a composite knowledge score of 50% may be different from the composite score 50% that study 2 reports. This is true because what was used to measure knowledge of HPV or HPV vaccination may be entirely different for study 1 and 2. Do the authors plan to use the specific proportions reported eg XX% reported that they had heard of of HPV vaccine in study 1 and then get a pooled proportion if this was reported in study 2? If this is what they plan to do, I think it should be made clearer in their protocol.

 Thank you for raising this issue. It is important that only comparable information is pooled. We will only be pooling comparable questions across studies. Data will first be categorized based on the type of data: categorical vs continuous and then further organized by the type of questions asked. For example, proportions for the number of correct responses for similar questions can be pooled or the average score on agreement with a statement (provided the questions and statements are similar). This approach has been used for COVID KAP meta-analyses.1,2,3 While many studies use common validated items from the World Health Organization (WHO) or the CDC, they often must make adaptations to fit the cultural or demographic context of the target population. 

Please see the updated text in the “Data synthesis” section, page 8, lines 15-23.

---

## [Editor Report · Decision Letter 1]

4 Nov 2024

Knowledge, attitudes, and practices towards Human Papilloma Virus and uptake of HPV vaccine: a protocol for a systematic review

PONE-D-24-23976R1

Dear Dr. Anni,

We’re pleased to inform you that your manuscript has been judged scientifically suitable for publication and will be formally accepted for publication once it meets all outstanding technical requirements.

Kind regards,

Hamidreza Karimi-Sari

Academic Editor

PLOS ONE
---

## [Editor Report · Acceptance letter]

15 Nov 2024

PONE-D-24-23976R1 

PLOS ONE

Dear Dr. Anni, 

I'm pleased to inform you that your manuscript has been deemed suitable for publication in PLOS ONE. Congratulations! Your manuscript is now being handed over to our production team.

Kind regards, 

on behalf of

Hamidreza Karimi-Sari 

Academic Editor

PLOS ONE